# Insights in Nutrition to Optimize Type 1 Diabetes Therapy

**DOI:** 10.3390/nu16213639

**Published:** 2024-10-26

**Authors:** Francesco Cadario

**Affiliations:** 1Division of Pediatrics, University del Piemonte Orientale, 28100 Novara, Italy; francesco.cadario@gmail.com; 2Diabetes Research Institute Federation, Miami, FL 33163, USA

**Keywords:** type 1 diabetes, type 2 diabetes, vitamin D, ω-6: ω-3 PUFA ratio, NAFLD, obesity, diabetes dyslipidemia, gluten-free diet, glycemic variability, ultra-processed food, personalized nutrition

## Abstract

Nutrition is an essential part of therapy for type 1 diabetes and is constantly evolving, offering growing opportunities to prevent this disease, slow down its evolution, and mitigate it. An attempt was made to bring together the current state of knowledge. In the path from the preclinical phase of the disease to its clinical onset, there is a phase known as the “honeymoon period” or partial remission, where different possible dietary options for combatting this disease have been presented. The most commonly used dietary models were compared, and the most frequent co-existing pathologies, such as overweight, non-alcoholic fatty liver disease, dyslipidemia, celiac disease, and metabolic instability, were addressed from their nutritional and dietary perspectives to provide clinicians with an updated framework of knowledge and support researchers in further investigations into the topic. Finally, a glimpse into the possible interplay between nutrition and the gut microbiome, food security, and ultra-processed food is provided. It is hoped that clinicians treating people with type 1 diabetes will be provided with further opportunities for the daily management of their patients through personalized nutrition.

## 1. Background

Given that type 1 diabetes develops in four different phases—(1) preclinical diabetes; (2) clinical presentation; (3) symptomatic outbreak, which is sometimes followed by temporary partial remission (or the honeymoon period); and (4) the chronic phase of dependency on insulin therapy—the nutritional and dietary approach to the disease should consider this progression over time [1]. Moreover, considering the increasing incidence of type 1 diabetes and younger ages at presentation, the identification of environmental determinants should be maximized and focused on in the first years of life because, if found, they should possibly be reversed [2]. Among the various environmental determinants considered are the nutritional ones, such as breast milk, vitamin D, and ω-3 polyunsaturated fatty acids (PUFAs), which have been supposed to protect against or counteract the progression of the disease; on the contrary, the use of cow milk rather than breastfeeding and the early introduction of gluten and coffee have been suggested to enhance the evolution of type 1 diabetes, though there are still no definite conclusions [3]. Until further studies have been carried out, breastfeeding must be encouraged, and the delayed introduction of nutrients must be carefully considered due to their roles in determining gut microbiota, and of possible players in the process of triggering autoimmunity [4]. Recent scientific investigations focused on associated pathologies, such as celiac disease, dyslipidemia, and overweight, with relevant implications for the reorganization of diet, thus adding further insights into the interplay between nutrition and type 1 diabetes. Moreover, the control of glycemic variability, which represents a major commitment for patients, might be reduced through nutrition and diet modifications, so they must be addressed in this work. Finally, the gut microbiota that elicit type 1 diabetes, food safety, and, in particular, the role of ultra-processed food, which occupies a growing space in the food choices of children and adults, must be considered in this review for their possibly causative role as environmental determinants of type 1 diabetes and the resulting metabolic deterioration.

## 2. Preclinical Diabetes

Environmental events causing type 1 diabetes should be identified within the first two years of age when the appearance of anti-β-cell islet autoantibodies peaks (more than one autoantibody defines the starting of the first stage of type 1 diabetes) [5]. So, within this period, in genetically exposed individuals, some infections elicit β-cell autoimmunity as the beginning cause. Viral infections may play this role, and Coxsackie B viruses are the most probable ones. Moreover, in observational studies of babies at risk, in those that would develop islet autoantibodies, raised glucose levels before the islet autoantibodies’ seroconversion were found [6]. Prof. A.-G. Ziegler hypothesized that an increase in glycemic concentration was consistent with a recent upper respiratory tract infection. The first appearance of anti-β-cell autoantibodies can regress without starting the disease [5]. Why this expression of autoimmunity can progress to more than one type of autoantibodies that represent the start of preclinical type 1 diabetes or, conversely, regress is unknown. It is certain that the different genetic backgrounds and levels of familiarity (i.e., the degree of kinship with other subjects with type 1 diabetes) differentiate the span of risk and are intertwined with possible environmental determinants that cause acceleration or delays. Definitively, the recurrence of type 1 diabetes in identical twins is nearly 50%, which means that, on the whole, non-genetic determinants account for an important weight in the risk of disease.

There is general accordance that the rising incidence of type 1 diabetes cannot be ascribed to genetics alone, and among causative environmental triggers are some related to nutrition or diet. Specifically, breastfeeding, cow milk, formula milk, gluten, probiotics, vitamin D, nicotinamide, and omega-3 have been investigated and found to be related to nutrition, birth weight, weight gain, BMI, childhood obesity, and gut microbiota [4]. To gain insight into the environmental determinants favoring type 1 diabetes, a model was offered by subjects undergoing pancreas transplantation. Such patients may experience the recurrence of autoimmune diabetes in the transplanted organ. In such a condition, a second onset of type 1 diabetes frequently occurs with the reappearance of autoantibodies against β-cells [7]. Preclinical studies in animal models suggested that vitamin D and ω-3 PUFAs are effective immune-modulatory agents of syngeneic and allogeneic islet transplantation [8]. In animal models of syngeneic islet transplantation, calcitriol (similar to vitamin D) appeared to play a role in the prevention of autoimmune recurrence. TNF-α is one of the major pro-inflammatory cytokines released upon the activation of Kupffer cells following intrahepatic islet transplantation, and calcitriol has been shown to significantly suppress TNF-α [9]. In addition, calcitriol may prevent β-cell apoptosis, which is a strong contributor to the loss of transplanted islets during the immediate post-transplant period.

Regarding nutritional deficits of vitamin D and their causality in type 1 diabetes in humans, notably, in a large series of Danish subjects, the level of 25(OH)D (which represents vitamin D status) at birth did not differ in subjects who developed type 1 diabetes within the 18th year of life [10]. Similarly, in a series of Italian children, no association was found between 25(OH)D levels at birth and risk of type 1 diabetes up to 10 years of age, except for a subgroup of migrant babies, who exhibited very low 25(OH) D levels (<2.14 ng/mL levels, OR of 14.02 (1.76–111.70) with respect to others) [11]. Notably, very low 25(OH)D levels are frequent in immigrant babies and mothers from North Africa in Italy and Europe [12,13,14].

Moreover, in a study of the European population using Mendelian randomization, genetic defects in vitamin D synthesis and exposure to type 1 diabetes were not significantly correlated [15]. In this study, the authors pointed out that the relationship was not applicable to non-European populations or for people with very low concentrations of vitamin D. On the contrary, in non-Caucasian people (i.e., among Egyptian children with type 1 diabetes), vitamin D receptor single-nucleotide polymorphisms causing low vitamin D status were associated with an increased risk of type 1 diabetes [16].

Today, there is a lack of interventional studies on immigrants in westernized countries that have replaced deep vitamin D deficiency and considered type 1 diabetes incidence as an outcome. Moreover, given the mounting evidence that a daily 2000–4000 IU of vitamin D supplementation for mothers sustains an adequate vitamin D status in breastfed babies, there is a need for strong supplementation in the planning of such a preventive strategy [17].

Regarding the ability of ω-3 PUFAs to prevent the preclinical phase of type 1 diabetes, a recently published study in two large Scandinavian cohorts found no associations between ω-3 PUFA intake during pregnancy and risk of type 1 diabetes in offspring, concluding that trials of eicosapentaenoic acid (EPA) and docosahexaenoic acid (DHA), the two most important ω-3-PUFAs in nutrition during pregnancy, should not be prioritized for the prevention of type 1 diabetes in offspring [18].

## 3. Presentation of Diabetes

The second phase of type 1 diabetes lasts from the stable appearance of at least two autoantibodies against β-cells to the clinical onset of the disease. It is characterized by dis-glycemia because of inadequate insulin production. Given the absence of clinical symptoms until the onset of the disease, this phase is habitually unrecognized. A variable span of time from the first years of life to the clinical onset of the disease, peaking at puberty, during which autoimmunity selectively reduces β-cell mass, should be the ideal period to counteract the process [19]. A corresponding rise in blood sugar may appear upon waking in the early morning or, likely, after meals in relation to reduced glucose tolerance. An inappropriate postprandial inhibition of glucagon-secreting α-cells strengthens hyperglycemia. Therefore, both insulin deficiency and a lack of suppression of glucagon in the post-oral glucose challenge concur with hyperglycemia [20]. Definitively, complex networks of islet cells and enteral cells producing incretins (e.g., glucagon-like peptide 1 (GLP-1), which is produced by L cells of the ileum/colon, and gastric inhibitory peptide (GIP), which is produced by the K cells of the duodenum) inappropriately concur to enhance hyperglycemia, which can be improved by nutritional intake, particularly by reducing glycemic load. In this complex scenario, current trials are in progress to cure and, secondarily, prevent type 1 diabetes through the administration of GLP-1 analog and dipeptidyl peptidase (DPP) inhibitors. An updated report on the topic was recently released [21]. In order to develop therapies that are available at pediatric ages, the possible nutritional support for resetting this network should be further investigated.

The process of reduction in the β-cell mass, which is characteristic of the second phase of type 1 diabetes, is progressive, not necessarily linear, and it has different speeds among individuals [22]. The environmental pressure of determinants might progressively reduce the age at the onset of childhood type 1 diabetes. The infiltration of T-cells (insulitis) is considered the hallmark of the progression of type 1 diabetes and can be detected predominantly in insulin-containing islets [23]. New image analysis methods revealed that T-cell density was higher in the exocrine and endocrine compartments of autoantibody-positive and type 1 diabetic individuals than in non-diabetic ones [23]. Given this picture of progressive insulitis, the surviving β-cells in Langherans islets producing insulin should potentially be preserved through correction of autoimmunity and by limiting inflammation, which can be achieved through vitamin D and ω-3 PUFAs. Particularly, it was found that the secondary prevention of T1D with calcitriol and paricalcitol is effective and safe if started soon enough after seroconversion [24]. Moreover, J.E. Löfvenborg et al. found that the combination of low PUFA status and high GAD65Ab (a kind of autoantibody related to type 1 diabetes) status interact positively with the progression of adult-onset diabetes (HR of 7.51 (95% CI 4.83, 11.69)), with evidence of an additive interaction in incidence (AP 0.25 (95% CI 0.05, 0.45)) [25]. However, in the same study, low intake of ω-3 PUFA in dairy was not associated with the incidence of diabetes [25]. An update about the possible regulation of immunocompetence by vitamin D and clinical investigation of ω-3 PUFAs for the amelioration of type 1 diabetes and inflammation in human and animal models were reported [26,27].

## 4. Clinical Onset and Presentation in the Partial Remission or Honeymoon Period

The third phase of type 1 diabetes is its clinical presentation, which is characterized by the classic symptoms of the disease and ketogenesis, leading to weight loss. Just after, sometimes, a temporary partial remission (or honeymoon period) may occur, which is usually in the first trimester of the symptomatic presentation of the disease. Since this remission represents the functional recovery of the share of persistent β-cells, which can be estimated as 10–30% of the pre-existing β-cell mass, it is important to preserve it because the residual β-cell function has been associated with a reduced incidence of hypoglycemia and microvascular complications in type 1 diabetes [19,28,29]. The level of C-peptide (the conjugation peptide of the α and β insulin chains) when fasting and after a mixed meal quantifies endogenous insulin secretion, and its level inversely is related to incident microvascular complications and the worsening of metabolic status in both type 1 and 2 diabetes [29,30,31,32]. Recently, the Innovative Approach towards Understanding and Arresting Type 1 Diabetes (INNODIA) consortium, by investigating a large series of patients during the first 12 months of disease, concluded that there were different age-related subtypes (or “endotypes”) of type 1 diabetes with different characteristics, among which the C-peptide levels referred to the age of onset of type 1 diabetes (lower in younger children) without significant age-related differences in the progressive rate of its decline [33].

In this phase of type 1 diabetes, insulin therapy plus nutritional support is crucial for driving the patient from ketoacidosis and hyperglycemia to euglycemia and, thus, preserving residual β-cells’ endogen function. Irrespective of the conclusions of the INNODIA study, hypothetically, the reversal of hyperglycemia could reduce the inflammation and antigenicity of β-cells targeted by T cells that are selective for the second-class HLA antigens DR and DQ. A long clinical history of symptomatic type 1 diabetes prior to exogenous insulin supplementation (i.e., initiation of therapy) and the concomitant ketoacidosis significantly reduce the likelihood of surviving residual insulin function and, thus, the consistency of remission. Thereby, the severity of diabetic ketoacidosis in children with new-onset type 1 diabetes was inversely correlated with insulin sensitivity [34]. Preclinical studies suggested that vitamin D and ω-3 PUFAs ameliorate autoimmunity in type 1 diabetes [26,27,35]. In this third phase of type 1 diabetes, the nutritional intake supporting insulin therapy leads patients from ketoacidosis to euglycemia and often reduces their insulin needs, introducing them to the honeymoon phase (conventionally referred to as a reduced insulin demand of <0.5 IU/kg/day, sharing a glycosylated hemoglobin percentage (HbA_1_c%), <7) and preserving residual β-cell endogen function. Although preclinical studies have suggested that vitamin D and ω-3 PUFAs are effective immunomodulatory actors, only a few studies in humans support this assumption.

In a randomized controlled trial, children and adolescent patients with new-onset type 1 diabetes were supplemented with vitamin D (cholecalciferol 140 IU/kg/day) or were not for one year. The former group had improved suppressor function in regulatory T cells and showed a slower decline of fasting C-peptide (as a marker of endogen insulin secretion). In the 12th month, a significantly lower insulin requirement was found in the supplemented group than in the control group [36].

In one cohort study, a whole type 1 diabetes series attending a single pediatric service was supplemented cholecalciferol plus sea-fish-derived ω-3 PUFAs (1000 IU/day plus 60 mg/kg/day) starting within the first trimester of the clinical onset of the disease [37]. A retrospective comparison with a previous series that received only cholecalciferol (1000 IU/day) showed that, one after the clinical onset of the disease, in the group co-supplemented with cholecalciferol and ω-3 PUFAs, the insulin demand was reduced (*p* < 0.01) and IDAA1c (insulin demand adjusted for HbA_1_c%; a composite index was calculated as IU/kg/day × 4 + HbA_1_c%, n. v. < 9) was reduced (*p* < 0.01), while HbA_1_c% did not differ (Figure 1).

Consistently, both studies pointed out the importance of optimizing the vitamin D level in type 1 diabetes, as well as in other autoimmune diseases. Achieving an optimal vitamin D status (25(OH)D levels 40–50 ng/mL) should allow the maximization of outcomes [38]. The potential benefit of the nutritional intake of vitamin D and ω-t PUFAs in the context of the honeymoon period of type 1 diabetes is further valorized because it has been shown to slow down the decline in pancreatic C-peptide secretion one year after clinical onset, whereas intensive diabetes management, including automated insulin delivery, even when achieving excellent glucose control, did not affect the rate of progressive decline of C-peptide [39].

Moreover, in adulthood diabetes, anecdotal experience of the remission of type 1 diabetes with the oral intake of inhibiting DPP-4 (100 mg of sitagliptin daily) and vitamin D (5000 IU of cholecalciferol daily) for four years was reported [40].

## 5. Symptomatic Type 1 Diabetes and the Chronic Phase of Dependency on Insulin Therapy

In the last phase of type 1 diabetes, given the dependency on insulin delivery, nutritional support becomes a cornerstone of therapy for patients; the insulin demand is calibrated to meals through the counting of carbohydrates and the correction of blood sugar levels according to insulin sensitivity. The use of this technology has simplified calculations for patients, and pumps using the continuous subcutaneous infusion (CSII) of genetically modified insulin (ensuring rapid bioavailability) have become the most reliable alternative to basal-bolus insulin therapy (through multiple daily injections, MDIs); they are assisted by continuous subcutaneous glucose monitoring (CGM) alone or in automated insulin delivery devices (closed-loop systems) [41]. In this scenario, the research on nutrition for achieving an ideal diet for patients with type 1 diabetes is in progress.

## 6. Statements on Nutrition in Type 1 Diabetes

There is a consensus among the scientific societies (ADA, EASD, and ISPAD) on nutrient intake [41,42,43]. The intake of carbohydrates, fats, and proteins is defined in Table 1. Fats should represent 30% of the total daily caloric content. The breakdown of fats should be <7–10% saturated fats, 10% monounsaturated fatty acids (MUFAs), and 10% polyunsaturated fatty acids (PUFAs). MUFAs are found in olive, sesame, and rapeseed oils, as well as nuts. Corn and sunflower are sources of ω-6 PUFAs, and fish, olive oil, and vegetables are sources of ω-3 PUFAs. Oily marine fish that are rich in ω-3 PUFAs are recommended once or twice weekly in amounts of 80–120 g. The guidelines recommend avoiding trans fatty acids as much as possible and limiting cholesterol intake to <200 mg/day. The daily protein requirement decreases from pediatric age (approximately 2 g/kg/day during childhood) to 1 g/kg/day for a 10-year-old child and to 0.8 to 0.9 g/kg/day in adolescence and adulthood. The fiber intake recommended in children is 14 g/1000 kcals (or, with an alternative formula, in children who are >2 years old, it is calculated as age in years + 5 = grams of fiber per day), up to 25–30 g/day, which represents the dose for adults.

Moreover, scientific societies recommend that diets be based on individual preferences, family and cultural background, and socioeconomic status. Consequently, one diet does not fit all, and a personalized diet should ensure better compliance and results. It is recommended to evaluate the distribution between macronutrients within a single meal to ensure the best postprandial metabolic compensation.

## 7. The Intake of Macronutrients and Its Effect on Postprandial Glycemia

Numerous studies have tested the effects of variable intakes of macronutrients on postprandial glycemia.

The critical point in balancing a diet, both when correcting nutritional errors and improving metabolic benefits, is substitution: replace what with what, and how much with how much of the other. Therefore, the following should be necessary: first, knowing what would be preferable for the individual patient, and second, referring to incoming evidence of different impacts of macronutrients on glycemic levels.

A comparison among four different isocaloric dietary regimes (high-carbohydrate; high-carbohydrate with extra fiber; low-carbohydrate, high-protein; low-carbohydrate, high-fat) for adults with type 1 diabetes was recently released [44]. This randomized crossover study concluded that the low-carbohydrate, high-protein meal showed the most favorable glycemic and metabolic profiles during the 4 h postprandial period. The indications from this important study are, however, much more detailed and not very exclusive; thus, readers are encouraged to refer to the article.

The difficulty in determining the effect of macronutrients on diabetes is related to the complex neuro-entero-hormonal imbalance that drives the blood sugar levels after meals.

Glucoregulatory hormones modulate gastric emptying, contributing to postprandial hyperglycemia, and in a dose-dependent manner, fat and protein regulate GLP-1, GIP, and glucagon release after meals. In two crossover euglycemic insulin clamp trials, 21 young patients with type 1 diabetes for >1 year received 30 g of carbohydrates and, alternatively, a low-protein or high-protein meal and a low-protein/low-fat or high-protein/high-fat meal. It was found that meals that were low in fat and protein had a minimal effect on GLP-1 and GIP, but there was an elevation of both after high-protein and high-fat meals. A glucagon rise was seen after high-protein vs. low-protein meals. In conclusion, the impact of fat and protein on postprandial glucose excursions may be mediated by the differential secretion of glucoregulatory hormones [45].

Moreover, the incoming evidence of biorhythms throughout the day influencing the effects of macronutrients on metabolism at breakfast, lunch, and dinner must be taken into account. Indeed, in adults with type 1 diabetes in a hybrid closed-loop system, nutritional factors other than the amount of carbohydrates were found to significantly influence postprandial blood glucose control [46]. These nutritional determinants vary among breakfast, lunch, and dinner, fixing the postprandial blood glucose profile.

In conclusion, today, optimizing the diet for type 1 diabetes is a hard challenge for dietitians, physicians, and patients.

## 8. Situations and Comorbidities

Frequently, comorbidities with type 1 diabetes or certain conditions may impose further specific diet approaches.

Diabetic ketoacidosis at onset or after impaired delivery of insulin needs particular attention. The management of diabetic ketoacidosis constitutes an important chapter in the treatment of type 1 diabetes that must be carefully evaluated by clinicians and is beyond the scope of this review. Accurate guidelines for nutritional intake have been reported, and the calibration of insulin delivery with carbohydrate intake per os or via IV is crucial [47].

Overweight in type 1 diabetes. Overweight in type 1 diabetes may be present from the onset or occur along the clinical course of the disease. Overweight more frequently characterizes type 2 diabetes but is not unusual in type 1 diabetes, given the increasing trend of overloaded nutritional caloric intake. An iatrogenic excess of insulin can also result in inadequate weight gain. The insulin resistance that adiposity entails in type 2 diabetes also works against people with type 1 diabetes; therefore, according to clinical studies on type 2 diabetes or conditions that cause risk for it, indications are also applicable for type 1 diabetes. Among the various dietary approaches proposed to reduce overweight in type 2 diabetes, the Mediterranean and low-carbohydrate diets are the most feasible for reducing weight in type 1 diabetes, keeping in mind that the main goal is to promote a caloric restriction without generating or limiting ketogenesis (see the ketogenic diet below).

## 9. The Mediterranean and Low-Carbohydrate Diets in Comparison: A Lesson from Type 2 Diabetes

The most commonly used diets for type 1 diabetes are the Mediterranean diet and low-carbohydrate diet, the metabolic impact of which was primarily addressed in type 2 diabetes and in overweight people. **The Mediterranean diet** is rich in whole grains, pasta, bread, whole wheat, eggs, poultry, fish, vegetables, legumes, fruits, and olive oil as the main ingredients, and it is low in red and processed meat according to the Mediterranean-style diet pyramid (macronutrient distribution: 50–55% carbohydrates, 15–20% protein, and 30% fat) [48]. The **low-carbohydrate diet** includes beef, veal, cold cuts, eggs, seasoned cheese, vegetables, and fruits (macronutrients: 30% carbohydrates, 30% protein, and 40% fat) [49].

Both the Mediterranean and low-carbohydrate diets are widely used in overweight non-diabetic adults, and they have become a reference for investigating insulin resistance and glucose homeostasis. In an open RCT, obese insulin-resistant adults at high risk of developing diabetes were randomly assigned to a Mediterranean or low-carbohydrate diet for 4 weeks, and the resulting average weight loss was 5%, which was 58% greater than that obtained when using the low-carbohydrate vs. Mediterranean diet [49]. Both diets showed similarly improved insulin resistance and fasting hyperinsulinemia.

In conclusion, the two diets are useful, and in a short time, the low-carbohydrate diet allows greater weight loss. However, cautions are necessary for prolonged treatment with a low-carbohydrate diet because, in a prospective interventional clinical trial of six months in adolescents with type 1 diabetes, the median intake levels of iron, calcium, vitamin B1, and folate significantly declined below the dietary reference intakes (DRIs) [50]. Notably, in this study, the favorable outcomes on weight and glycosylated hemoglobin (HbA_1_c) with the low-carbohydrate diet were related to the decreased intake of highly processed food (see below, nutrition safety and ultra-processed food).

In a recent study of type 1 diabetic children, adherence to the Mediterranean diet was found to gain improvements in HbA_1_c% and time in range (TIR, the percentage of time spent with the blood sugar target of 70–180 mg/dL) [51].

**Early time-restricted carbohydrate consumption** is an innovative diet that aims to limit the intake of carbohydrate-rich food in the morning and early afternoon to comply with circadian variations in glucose tolerance. This diet was tested in type 2 diabetes and obese adults in comparison with a Mediterranean-style diet in an open-label RCT that was 12 weeks long. It produced comparable reductions in weight and fat mass and similar improvements in HbA1c, insulin resistance, glucose tolerance, and glucose variability [52]. This diet is available for reductions in the weight of adults with type 2 diabetes and obesity, and it has similar benefits to those of the Mediterranean diet.

**Low ω-6 to ω-3 polyunsaturated fatty acid ratio diet.** This diet, which was carefully investigated in 17 adolescent subjects with non-alcoholic fatty liver disease (NAFLD), overweight, and type 2 diabetes, provided an interesting insight into the impact of a low ω-6 to ω-3 PUFA ratio on insulin clearance both during fasting and after meals [53]. The main result was an enhanced clearance of insulin, regardless of weight reduction, and other dietary changes in an isocaloric regimen. Notably, after three months of diet, the best results in terms of HbA_1_c and insulinemia reductions were found in subjects with worse glucometabolic control at enrollment. So, the low ω-6 to ω-3 PUFA ratio diet highlights a specific pathway related to a low ω-6 to ω-3 PUFA ratio, which, by itself, improves insulin clearance in NAFLD patients, making it potentially useful in type 2 diabetes, obese people, and, possibly, type 1 diabetes. In animal models, fish oil supplementation provided a similar effect mediated by improved skeletal muscle mitochondrial function [54].

## 10. The Ketogenic Diet

There is growing interest in the ketogenic diet for diabetes, as is clear from a growing number of scientific articles on the subject [55], especially for type 2 diabetes (Figure 2).

In summary, a strong limitation of carbohydrates is proposed (approximately 8% of the daily caloric intake), in addition to the complete abolishment of starches and limitation of simple sugars, especially those in highly processed foods, allowing only small amounts of fruit. Vegetables are indicated as a source of ω-3 PUFAs, and avocado, olives, nuts, olive oil, cream, or butter are sources of fat. Meat, fish, eggs, and dairy products can be put into the everyday menu as a source of protein. The percentage of fats increases, usually to 70–80%, and protein frequently accounts for about 20% of the energy share. A caloric restriction adds effectiveness when the patient is overweight.

When used in type 2 diabetes, the ketogenic diet results in reductions in glucose and insulin concentrations in serum, an improvement in HbA_1_c, and a reduction in the HOMA-IR (an indicator of insulin resistance). The use of the ketogenic diet in type 1 diabetes began in 2006 with a 4-year-old girl with resistant epilepsy syndrome, and the seizure disorder was treated with a ketogenic diet. In this patient, a clear benefit in terms of convulsions was accompanied by unexpected metabolic advantages [56]. Nowadays, the use of a ketogenic diet in type 1 diabetes is supported by a limited number of studies (Figure 2). Significant metabolic benefits have been reported, including better blood sugar control, reduction in HbA_1_c, and reduced daily insulin demand. The lipid profile results in a decrease in triglycerides. The recurrence of hypoglycemia and the possible onset of diabetic ketoacidosis require patients to be carefully supervised. Physiological ketosis is characterized by a slight concentration of ketone bodies within the range of 0.5–3 mmol/L, without a reduction in the blood pH value. In a systematic review, the authors concluded that further research is needed in this field, especially studies with a long follow-up period [55].

The ketogenic diet shows similarities with other diets, such as the Mediterranean, low-carbohydrate, and low ω-6 to ω-3 PUFA ratio diets, but indubitably, its task is shifting the main source of energy supply from glucose to ketones. Future research should investigate both the metabolic aspect and immunological one, since insulin requirements are preserved or reduced over time. It was hypothesized that butyrate and lactate (ketone bodies) possibly mediate the gut microbiota with an increase in the number of *Bacteroides* organisms and the anti-inflammatory action of a reduced ω-6 to ω-3 PUFA ratio. The main goal of the ketogenic diet is preserving endogenous insulin secretion, which correlates with reduced risk of complications over time [28].

## 11. The Diet in Type 1 Diabetes and Dyslipidemia

Type 1 diabetes affects both glucose and lipid metabolism, leading to suggestions that diabetes should be named “lipidus” rather than “mellitus” [57]. Indeed, in the preclinical phase of type 1 diabetes, insulin deficiency deregulates free fatty acid (NEFA) suppression after the meal glucose load, causing hyperchylomicronemia. It is this, when the inflow becomes excessive, that confers a “milky” appearance of whey; once upon a time, this was recognized as a characteristic of diabetes (and a preferable diagnostic criterion for physicians rather than the sweet taste of urine). This milk-like serum was feared by clinicians as a life-threatening event for the patient due to possible fatal cardiotoxicity, as in ketoacidosis, hyperglycemia, and severe dehydration.

Today, hypertriglyceridemia represents the hallmark of diabetic dyslipidemia, and fibrates are largely used to reduce the risk of consequently possible acute pancreatitis. Hypertriglyceridemia concurs with the hepatic production of VLDL, which constitutes the major transport vehicle of triglycerides (with Apo B100 lipoprotein as the carrier) by approximately 50% and the intestinal synthesis of chylomicrons (with Apo B48 lipoprotein), mainly in the postprandial phase, by 50% due to a rearrangement of the excessive intestinal absorption of cholesterol [58]. There is evidence of atherogenic properties of triglyceride-rich lipoproteins. Postprandial hyper-chylomicronemia that is assembled in the intestine through disturbance in cholesterol absorption is a major player in the atherogenicity of diabetic dyslipidemia [58].

In this complex scenario, insulin therapy regulates the metabolic imbalance, supported by the nutritional limitation of the glycemic load, cholesterol, and triglycerides in the diet. In an Italian pediatric type 1 diabetes series showing a suboptimal lipid profile, six-month structured training in the Mediterranean diet by a dietician and an increased intake of fiber (the mean assumption was 18.0 ± 0.4 g/day) resulted in decreased cholesterol intake (*p* < 0.0001), increased fiber intake (*p* < 0.0001), and decreased production (*p* < 0.001) of LDL-c, non-HDL-c, and the total cholesterol–HDL-c ratio independently of weight decrease and irrespectively of the glucose control [59]. In type 1 diabetes patients, even with good glycemic control, qualitative and functional abnormalities in lipoproteins were found. Although the mechanisms underlying diabetic dyslipidemia remain partially unclear, subcutaneous insulin administration and subsequent peripheral hyperinsulinemia might represent an important determinant [60].

The use of statins is the usual therapy for diabetic dyslipidemia in adults, and nowadays, atherogenic risk in type 1 diabetes children leads to the hypothesis of their use in pediatrics for tighter control of dyslipidemia, such as in familial hypercholesterolemia [61]. Innovative therapies are envisaged, including such nutritional therapies as icosapent ethyl (a highly purified EPA), in type 2 diabetes [62,63]. Today, in type 1 diabetic individuals aged ≤30 years without evidence of vascular damage and microalbuminuria, it seems reasonable to delay statin therapy until the age of 30. Below this age, statin therapy should be managed on a case-by-case basis [64] (Chapter 9.4.3). Until validated therapies at pediatric ages are developed, the administration of an ω-3 PUFA supplement derived from fish oil and ultrafiltered from heavy metals is safe and results in a reduction in triglycerides, and an ω-6:ω-3 ratio ≤ 3 improves metabolic control [65].

## 12. The Gluten-Free Diet in Type 1 Diabetes and Celiac Disease

Type 1 diabetes clusters with other autoimmune diseases, such as Hashimoto’s thyroiditis, Addison disease, pernicious anemia, atrophic gastritis, vitiligo, and celiac disease. The latter has specific and strong dietary implications because it requires the accurate exclusion of gluten-containing foods, such as wheat, rye, barley, and oats. This diet needs to be followed strictly and over one’s entire life. The implication of the coexistence of diabetes and celiac disease makes planning diets difficult; moreover, there are metabolic and immune concerns, which were well summarized in a recent review [66].

Given that gluten-free foods have a high glycemic index, and the gluten-free diet could lead to better intestinal absorption, at the start of the diet, the glycemic load is increased with meals and insulin demand, often resulting in weight gain in patients. So, patients with co-existing conditions must be managed by experienced dietitians in close collaboration with diabetologists. The choice of foods with a low glycemic index may be vital for counteracting such unfavorable sequelae from gluten-free diets. A gluten-free diet does not eliminate the differences between celiac disease and non-celiac disease. In a study of young people with type 1 diabetes—10 with co-existing celiac disease vs. 7 without—those with celiac disease showed a shorter time-to-peak blood glucose level (77 vs. 89 min, *p* = 0.03), a higher peak (167 vs. 131 mg/dL, *p* = 0.001), and higher postprandial blood glucose levels than those in type 1 diabetes patients without celiac disease (151 vs. 126 mg/dL, *p* = 0.01) despite their pre-meal blood glucose levels being similar (165 vs. 155 mg/mL, *p* = 0.28) [67]. Naturally gluten-free foods, such as rice and corn, could be better accepted, and their glycemic index could be mitigated by cooking methods and the use of condiments to slow down the absorption of carbohydrates [68]. The limitations of dietary choices and weight gain often reduce patients’ compliance with gluten-free regimens. So, dietary non-adherence to gluten-free diets in type 1 diabetes patients with celiac disease is frequent; thus, to improve compliance, the possible benefits in terms of metabolic control, reduction in hypoglycemia, and decreased long-term complications should be presented to patients.

A peculiar characteristic of celiac disease associated with type 1 diabetes is the frequent absence of gastroenteric symptoms. Only approximately 10% of patients present specific symptoms. Thus, the ISPED and ISPAD recommend looking for the presence of specific autoantibodies (transglutaminase, as a marker of celiac disease) at the diagnosis of type 1 diabetes and every 2–2.5 years within the first five years of the disease, even in the absence of clinical symptoms for detecting possibly silent celiac disease [69].

Beyond the common genetic background of the two diseases linked to the HLA DQ2 and DQ8 alleles, there are causative relationships between the two pathologies. Both show an increasing trend. Co-occurrence is greater than that expected according to genetic risk, meaning that environmental determinants contribute to the increase. Altered intestinal permeability and microbiota have been hypothesized to play this role [70]. There are clinical and immunological improvements in type 1 diabetes resulting from gluten-free diets. An anecdotal case was reported of a 6-year-old child with type 1 diabetes who had undertaken a gluten-free diet since the clinical onset of diabetes, even in the absence of positive antibodies for celiac disease, and this child experienced persistent total remission that lasted for twenty months [71]. Another study described 15 children with new-onset type 1 diabetes who were introduced to a gluten-free diet after diagnosis, regardless of celiac disease, and they showed more persistent remission (C-peptide > 300 pmol/L after a mixed breakfast meal) 12 months from onset than that of two previous cohorts (*p* < 0.001). [72].

In observational studies of subjects at risk for diabetes, the development of autoantibody markers for type 1 diabetes usually but not always preceded the appearance of anti-tissue transglutaminase autoantibodies [73]. In Italy, a diagnostic program for the preclinical identification of type 1 diabetes and celiac disease in the pediatric population through the detection of autoantibodies of type 1 diabetes and celiac disease started in 2024 [74]. It is expected that this study will provide new data on the interactions between the two diseases.

In conclusion to the question by E. Klieverik in his article, a gluten free diet is a promising approach to be considered in nutrition for type 1 diabetes [66].

## 13. The Diet and Gut Microbiota in Type 1 Diabetes

New therapeutic options for nutrition in type 1 diabetes through diets are oriented toward improving the gut microbiota.

Gut microbiota transplant therapy in type 1 diabetes, which is supported by a solid scientific literature, is a promising therapy [75]. While experimental trials are in progress, today, it is obvious that an attempt should be made to “foster” the growth of the most favorable gut microbiota to delay the onset or progression of the disease and improve insulin therapy.

Dietary components have profound effects on the direction of gut microbiota. The “Western” and “Mediterranean” diets are the most commonly used dietary models of reference, and they shift the microbial composition of microbiota; through bacterial metabolites, they may enhance or counteract systemic inflammation and metabolic endotoxemia [76]. This well-established pathway has shown a favorable impact on the prevention and treatment of type 2 diabetes and obesity. This is not equally transferable to type 1 diabetes. The involvement of the gut microbiota as protective or favoring the pathogenesis of type 1 diabetes pathogenesis is recognized, and, in particular, the protective role of microbially produced short-chain fatty acids (SCFAs) from type 1 diabetes was reported. In the Innovative Approaches to Understanding and Arresting Type 1 Diabetes (INNODIA) study, the microbiota in type 1 diabetes were recently addressed [77]. During a 2-year follow-up of 98 newly diagnosed type 1 diabetes patients vs. 194 unaffected family members who were autoantibody-positive and at risk for the disease, there was a longitudinal increase in 21 different bacterial species. The relative abundance of *Faecalibacterium prausnitzii* was inversely associated with the HbA_1_c levels at diagnosis. Individuals with a rapid decrease in C-peptide levels in the newly diagnosed type 1 diabetes group had lower gut microbiota diversity than that of the others. In the unaffected group, nineteen individuals who developed overt disease within the follow-up period were found to have increased abundance of *Sutterella* sp. *KLE1602* in comparison with other undiagnosed individuals [77].

In an Italian study on 40 children with type 1 diabetes and 56 healthy controls in a multivariate analysis of risk, after adjusting for confounding factors, the likelihood of having diabetes was significantly higher in those with a larger amount of Firmicutes, a smaller amount of *Bifidobacterium* spp., and a larger total carbohydrate intake [78]. However, in the largest series of type 1 diabetes examined, the microbiota was correlated with α-diversity (local diversity), but this was not consistent across different geographical areas (δ-diversity). Moreover, nutritional behavior characteristics, such caloric intake, extra portion requests, and meal distribution during the day, had an interplay with gut microbiota. In another Italian case, a control study of 10 children with β-cell autoimmunity who were at risk for type 1 diabetes and 10 healthy children, increased intestinal permeability and differences in gut microbiota composition were contemporaneously documented before the onset of diabetes in cases vs. the healthy controls [79].

The Environmental Determinants of Diabetes in the Young (TEDDY) study, a large and detailed longitudinal functional profile of the developing GM related to islet autoimmunity, supports the protective effects of SCFAs in early-onset human type 1 diabetes [80]. In an animal disease model of NOD mice, an inverse relationship of the blood and fecal concentrations of the microbial metabolite acetate and butyrate SCFAs with key markers of type 1 diabetes progression was found. Acetate decreased the frequency of autoreactive T cells in lymphoid tissues, limiting B cells and their ability to expand autoreactive T cells, whereas a butyrate-yielding diet enhanced gut integrity and decreased serum concentration of diabetogenic cytokines such as IL-21 [81].

Bell, K.J. and coauthors addressed a single-arm trial in adults with long-standing type 1 diabetes and administered a supplement for 6 weeks with amylose maize-resistant starch modified with acetate and butyrate (HAMSAB) [82]. In a 12 weeks follow-up, changes in the gut microbiota were assessed. Increased concentrations of SCFAs acetate, propionate, and butyrate were found in stools and plasma, in concert with a shift in the composition of the GM. A diet with amylose maize-resistant starch modified with acetate and butyrate (HAMSAB) was administered for 6 weeks in 25 patients, and a 12-week follow-up was performed to evaluate changes in the gut microbiota over time. A shift in the composition of gut microbiota and increased concentrations of SCFAs (acetate, propionate, butyrate) in stools and plasma were found. The subjects with the highest SCFA concentrations showed the best glycemic control even when their glucose control and insulin requirements did not change. The *Bifidobacterium longum* and *Bifidobacterium adolescentis* concentrations were correlated with lower HbA_1_c and decreased basal insulin requirements and more circulating B and T cells with the regulatory phenotype. The persistence of these effects beyond HAMSAB administration suggested that targeting dietary SCFAs may alter immune profiles, promote immune tolerance, and improve glycemic control in type 1 diabetes treatment [82].

Some deductions seem to emerge overall; the gut microbiota play a protective or favoring role in the pathogenesis of diabetes; the SCFAs in the gut show protective effects against early-onset type 1 diabetes; and specific foods can increase the SCFA concentration in the gut [77,78,79,80,81,82].

## 14. Diet for Improving Glycemic Variability in Type 1 Diabetes

A major problem for type 1 diabetes patients, and that which is most challenging for diabetologists, families, and caregivers, is limiting glycemic variability (GV).

When GV involves wide and repeated oscillations with frequent high hyperglycemia and severe hypoglycemia, which are apparently uncorrectable, we speak of unstable diabetes or, rather, “brittle diabetes”, indicating fragility and meaning that there is both a precarious metabolic balance and a scared person, with psychological and physical implications. This complex problem is well addressed in the specific literature and would be beyond the scope of this article [83].

Glucose variability habitually occurs in most type 1 diabetic persons, limiting their skills in school and at work and the freedom to practice physical activity and sports, with important psychological setbacks and worsening of the quality of life.

In “brittle diabetes” and in GV, there is an important nutritional share, which, beyond the enormous advances in insulin pump therapy (CSII) and real-time interstitial glucose monitoring (CGM) with interconnected automated systems, also requires dietary supply.

Given that subcutaneous insulin administration bypasses the hepatic filter, which physiologically halves its concentration, the adequate dosage of boluses at meals often can be addressed with a diet that limits postprandial hyperglycemia.

Taking this path without upsetting patients’ diets and habits could be useful for restoring the patient–physician relationship, resetting the centrality of the patient’s food choices, overcoming psychological frailty, and possibly reducing metabolic instability.

An accurate review of insulin secretion and the complex neurohormonal regulation of glycemia in physiology and diabetes was reported in the review of G.D. Dimitriadis and coauthors [84]. According to this article, “the regulation of postprandial glucose handling is extremely complex, and the numerous factors involved do not operate individually; rather, they function in concert, often promoting the simultaneous secretion of their antagonists to balance their effects”. For this reason, it is inseparable, on the one hand, to promote the best distribution of macronutrients in the diet and, on the other, to possibly reset the intestinal neuro-endocrine balance with specific nutrients.

Herein are reported some conclusions from two studies conducted among young Australian type 1 diabetes patients. In the first, nutrition was addressed using CGM records in terms of time-to-peak glycemic responses (TTP) after consuming meals under controlled or free-living conditions at breakfast and dinner [85]. The TTP variability was found to be greater within a person than the variability between persons for all meal types.

In the second study, in two crossover euglycemic insulin clamp clinical trials, it was found that the impacts of fat and protein on postprandial glucose excursions were probably mediated by the differential secretion of the glucoregulatory hormones GPL-1, GIP, and glucagon [45]. In conclusion, controlling glucose excursions is challenging for patients, families, and physicians because individual differences in glycemic responses to meals must be addressed; moreover, the variability within individual patients must be accounted for. Therefore, reducing GV through dietary components is one of diabetologist’s major commitments.

Of note, in a study of Italian adults with normal or reduced glucose tolerance or type 2 diabetes, the GV was reduced with the intake of small amounts of lipids or proteins before meals. Interestingly, the plasma glucose rise induced by OGTT was lower across classes of glucose tolerance (normal glucose tolerance: −32%, impaired glucose tolerance: −37%, and type 2 diabetes: −49%; *p* < 0.002) [86]. The authors concluded that the mild stimulation of GLP-1 and GIP was associated with a slower exogenous glucose rate increase, improved β-cell function (as these were type 2 diabetes patients), and reduced insulin clearance after OGTT.

In another Italian study, eleven adults with type 1 diabetes on an insulin pump underwent a week-long study with a randomized cross-over design, alternatively adding monounsaturated fat (extra-virgin olive oil), saturated fat (butter), or a low-fat diet to meals [87]. It was found that, with the addition of extra-virgin olive oil, there was a significant decrease in the blood glucose area under the curve in comparison with the addition of butter or a low-fat diet. Postprandial GLP-1 and triglyceride increased with the addition extra-virgin olive oil in comparison with the addition of butter or a low-fat diet.

Finally, it was found in a single anecdotal case of a type 1 diabetic adolescent Italian girl [88] that the daily supplementation of ω-3 PUFAs (50 mg/kg/day, EPA 66% DHA 33%) in a Mediterranean-style diet produced stable metabolic benefits. This resulted in a reduction in the mean subcutaneous glucose (*p* < 0.0001), a reduction in GV (*p* < 0.07), and an increased time in range (TIR) 70–180 mg/dL (*p* < 0.0001) in comparison with an equivalent period prior to ω-3 PUFA supplementation. The reductions in subcutaneous glycemia and glucose variability in this girl were documented even during prolonged sportive activity, without an increase in hypoglycemia. No side effects were found, and triglycerides persisted in the normal range.

In conclusion, glycemic variability may be addressed with the addition of proteins (one boiled egg), lipids (g 50 of parmesan cheese), MUFAs (monounsaturated fatty acids, olive oil), or fish oil supplementation (ω-3 PUFAs at 50 mg/kg/day) assumed before or within the meals, and the post-meal increase is likely related to glucoregulatory hormones such as GLP-1 reducing gastric emptying and glucagon. TTP and TIR probably represent easier parameters when looking for improvements in daily glycemic control in many patients with type 1 diabetes [86,87,88].

## 15. Personalized Diet in Type 1 Diabetes

The above-mentioned anecdotical case of a sportive teenage girl with type 1 diabetes is representative of the potentiality of a personalized diet in nutrition for type 1 diabetes [88].

In summary, a girl with the onset of the disease at 9 years of age, who was on MDI therapy assisted by CGM, began supplementation with ω-3 PUFAs derived from fish oil (50 mg/kg weight/day, EPA/DHA, 66%/33%) at 11 years and 6 months of age due to the finding of a high AA/EPA ratio (arachidonic acid ω-6 (AA)///eicosapentaenoic acid ω-3 (EPA) ratio, 14.19), with metabolic benefits in free-living and sporting activities. The reduction in AA/EPA (2.27) that was in the normal range (n.v. 2–2.5) confirmed the adequate intake of ω-3 PUFAs. After a year and a half, she (13 years old) voluntarily suspended the supplementation, with worsening clinical and metabolic outcomes. The CGM records allowed the comparison of data between equivalent periods prior to ω-3 PUFA supplementation and after, and both were one year and a half long. Over this three-year period, 344,020 valid data points on subcutaneous glucose levels were collected [88]. An interpretation was performed according to the 80–180 TIR, which motivated the patient to restart the supplementation (Figure 3). Even during comparable prolonged sporting activities, a reduction in blood sugar levels and CV was demonstrated (Figure 4). Unexpectedly, a reduction in the average blood sugar levels in fasting time was found; in the context of a substantial stable insulin demand, this highlighted the increased insulin sensitivity in fasting with ω-3 PUFA supplementation (Figure 5). The analytical examination of the data clearly demonstrated, first, to the girl, the opportunity to restart supplementation with omega-3 PUFAs and, for clinicians, the potentiality of a personalized approach to nutrition in type 1 diabetes.

## 16. Planning Personalized Nutrition

One of the major commitments in patient-centered care is identifying nutritional interventions that are beneficial to a specific patient. Medical nutrition therapy that is planned for an individual person is called personalized nutrition (PN) and is part of precision medical therapy [89]. The PN approach, which is usually stigmatized by the statement “one diet does not fit all”, even without evidence from randomized controlled trials (RCTs), must necessarily be based on statistical evidence. Engaging this type of investigation in type 1 diabetes shows strengths and weaknesses that need to be kept in mind.

Metabolomics, which is the study of metabolites of a specific nutrient, could improve the methodological aspect because it can bypass the measurement of exposure to specific dietary components—for example, by using questionnaires on frequency and doses with which certain foods are eaten, which can often be misleading. Given that a nutrient generates specific metabolites that can be measured in the blood (or in the urine), it could be ascertained whether a person ate specific nutrients by finding derivative biomarkers of their consumption [90]. Moreover, this strategy could overcome the problem of choosing the optimal dose of nutrient to be administered, since derived metabolites would be more easily quantified and optimized into a specific target range. This is essential when choosing metabolites to consider interactions with other components of a diet that use common metabolic pathways or have opposing effects (e.g., ω-3 and ω-6 PUFAs, which show opposing effects on inflammation, and saturated fats, which compete with PUFAs for common metabolic pathways). Therefore, the use of the ratio between bioactive agonist and antagonist metabolites is more descriptive than that of agonists alone (e.g., the ω-6:ω-3 PUFAs ratio rather than ω-6 alone) [65]. Finally, ethical concerns may arise in case vs. control RCTs with respect to keeping someone in the control group in a known nutritional deficit for the investigation time.

In type 1 diabetes, the technologies of CGM and CSII generate a large amount of data on individual patients, and these are measured outside of the clinic and are deliverable via electronic devices; these could turn out to be useful for effective investigations centered on single patients through the application *n*-of-1 trials [91].

Pending reliable RCTs or crossover studies, which represent the gold standard of research, PN could promote an extraordinary opportunity for single patients if (i) the nutrients administered replace verified deficiencies, (ii) previous research supports the hypothesis of their pathogenetic role in the disease, (iii) nutrient concentrations (or their biomarkers) measured at the start and endpoint in *n*-of-1 trials were under/in the ideal range, and (iv) clinical outcomes can be quantified and comparable before/after nutrient exposure.

Moreover, possible enzymatic defects could be considered on a genetic basis (e.g., Δ-6 desaturase) in terms of their interference in the path from a given nutrient (e.g., α-Linoleic acid 18:3 ω-3 PUFA) that limits the synthesis of metabolites at the derived levels (e.g., EPA 20:5 ω-3 and DHA 22:6 ω-3 PUFAs), so there may be a discrepancy in the intake of a nutrient (identified using questionnaires on nutritional intakes) without a corresponding increase in metabolomic-based investigations [65].

In *n*-of-1 trials, the planning of the randomization of treatment periods is crucial [91]. The AABB treatment sequence does not protect against confounders whose effect could intervene in the outcome over time (e.g., intercurrent puberty in the above-mentioned case report). The paired ABAB design should share a more appropriate timing (e.g., in the specific case, the worsening of the parameters under observation with the suspension of supplementation). Notably, even if an *n*-of-1 trial provides decisions for an individual patient, it can likely be useful for other similar *n*-of-1 trials. This mimics how physicians learn from their previous clinical experience and from the clinical experience of colleagues [91]. In conclusion, from the above statement about personalized diets in type 1 diabetes, “a PN could be suitable for much more than just one person”.

## 17. Food Safety and Ultra-Processed Food

Food safety has strong connections with the everyday diet of children and adults, and, of course, it concerns both type 1 and 2 diabetes, overweight, obesity, and non-communicable diseases in humans.

Herein, according to the topic of this review on nutrition and dieting for type 1 diabetes, we focused on the degree of food processing. There is mounting evidence that ultra-processed foods (UPFs) worsen type 1 diabetes and, with the co-occurrence of overweight and reduced physical activity, add the risk of arterial hypertension, cardiovascular disease, and mental depression in people with type 1 diabetes.

UPFs are defined as industrial formulations made by breaking down whole foods into food-derived substances (e.g., fats, sugars, starch, isolated proteins) and altering and re-combining them with additives, such as colors, flavors, and emulsifiers in products [92,93]. Typical examples of UPFs are soft drinks, fast foods, chicken nuggets, instant soups, fruit drinks, and flavored yogurts. UPFs are produced and sold by multinational food companies and are convenient, tasty, and hyper-palatable [94]. These foods replace fresh, minimally processed foods, fruit, vegetables, and freshly prepared meals and have displaced a significant portion of the traditional diets of various populations [94]. Recently, an umbrella review of existing meta-analyses on health outcomes of UPFs was released [95]. It was found that greater exposure to UPFs is linked with a higher mortal risk of incident cardiovascular disease, type 2 diabetes, anxiety, common combined mental disorders, adverse sleep outcomes, wheezing, and obesity.

The metabolic outcomes of diets rich in UPFs were addressed in a study based on a dataset of 98 ready-to-eat foods [96]. Foods were clustered within three processing groups based on the international NOVA classification: (1) raw and minimally processed foods; (2) processed foods; (3) ultra-processed foods [92,93]. The degree of food processing was correlated with the satiety index, glycemic index, and glycemic glucose equivalent (GGE). The latter indicated the amount of glucose that would have an effect equivalent to that of a certain food. This new index allowed the identification of the quantity of food affecting the blood glucose level. In this study, there were strong correlations among the GGE, satiety index, and degree of food processing, while the glycemic index was not correlated with the degree of processing [96]. Thus, the more food is processed, the higher the glycemic response and the lower its potential satiety; conversely, minimally processed foods are more satiating and less hyperglycemic than UPFs [96].

In a cross-sectional study, 120 children and adolescents with type 1 diabetes in Rio de Janeiro underwent an assessment of their food consumption using 24 h food recall in 2015, and they were classified into five dietary patterns for obesity and metabolic control (HbA_1_c) [97]. Among the five dietary patterns identified, the pattern characterized by a greater consumption of UPFs had a higher odds ratio (OR) for higher HbA_1_c levels (OR = 3.49; 95% CI = 1.18–11.16), allowing the conclusion that higher consumption of UPFs can be detrimental to glycemic control in children and adolescents.

Recently, UPF consumption and the risk of diabetes were addressed in the Atherosclerosis Risk in Communities (ARIC) study [98]. In a community-based cohort of 13,172 middle-aged adults in the USA, the frequency of food assumption was matched, and food was categorized by the processing level (according to the NOVA classification system). It was found in a follow-up over 21 years that there were 4539 cases of incident diabetes. Participants in the highest quartile of UPF portion consumption (8.4 servings/day) had a higher risk of diabetes than that of those in the lower quartile (HR 1.13; 95% CI 1.03, 1.23). Each serving consumed daily was associated with a 2% higher (HR 1.02; 95% CI 1.00, 1.04) risk of diabetes. This higher risk of incident diabetes included sugary and artificially sweetened beverages (29%), ultra-processed meats (21%), and sugary snacks (16%).

## 18. Conclusions

An attempt was made in this review to summarize the current knowledge on nutritional therapy for type 1 diabetes as a starting point for future research.

In conclusion, until robust scientific evidence is provided, today, it seems that some key points can be reported to patients and parents when they ask how to improve their diet.The prevention of type 1 diabetes is elusive. Breastfeeding must be recommended to endorse the superiority of the gut microbiota induced by breast milk in the window period of the first years of life, and the early introduction of solid nutrients, gluten, and cow milk must not be recommended [4]. Vitamin D can be supplemented in non-Caucasian populations who are at risk for severe deficiency and should be offered to new breastfeeding mothers.The consolidated differences among people with type 1 diabetes produce increasing abilities of physicians and dieticians to identify subtypes (or endotypes) that are useful in the path toward personalized nutritional therapy [99]. Today, for newly diagnosed patients, dietary indications and insulin therapy based on personalized nutrition should help correct their documented nutritional defects, vitamin D deficiency, and high ω-6:ω-3 PUFA ratios.Incoming socioeconomic disparities and the spread of non-traditional dietary patterns constitute an important challenge for people with type 1 diabetes due to their worsening metabolic instability. Perceived glucose variability and hyperglycemia are the strongest predictors of daily diabetes distress [100], so they must be addressed through nutritional education and psychological support to individual patients.

Finally, given that the doctor-patient relationship is an essential tool for therapeutic success, further possibilities derive from the individual experience of clinicians, of which some are reported in Appendix A.

## Figures and Tables

**Figure 1 nutrients-16-03639-f001:**
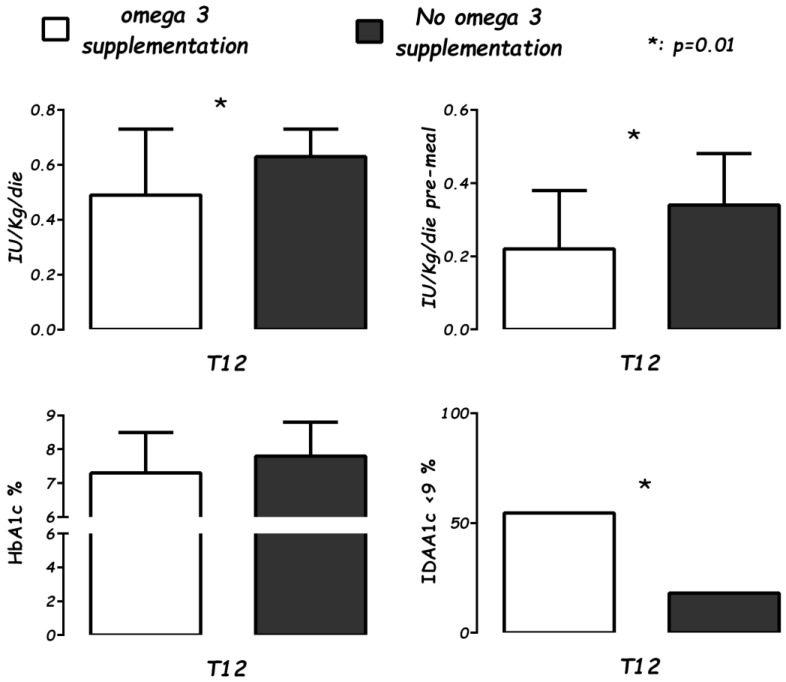
In a study with a cohort design, all type 1 diabetes onsets in a single center consecutively received ω-3 PUFA supplementation (60 mg/kg/day, EPA 66%, DHA 33%) for one year and were compared with a whole series of onsets in the previous two years without supplements. In the 12th month after onset, the daily insulin needs, pre-meal boluses, HbA1c%, and IDAA1c (the insulin dose adjusted for glycosylated hemoglobin percentage) as a surrogate for the partial remission index (≤9) were compared in cases (22 children, white bar) and in controls (37 children, black bar) [37].

**Figure 2 nutrients-16-03639-f002:**
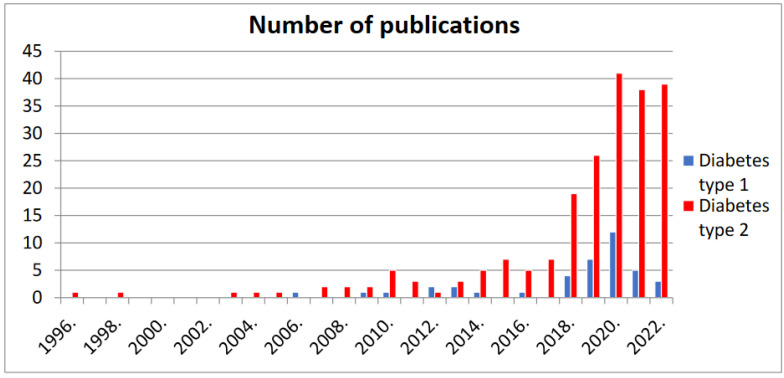
Comparison of the number of publications in PubMed for type 1 diabetes and type 2 diabetes related to The Ketogenic Diet on the Prophylaxis and Treatment of Diabetes Mellitus: A Review of the Meta-Analyses and Clinical Trials [55].

**Figure 3 nutrients-16-03639-f003:**
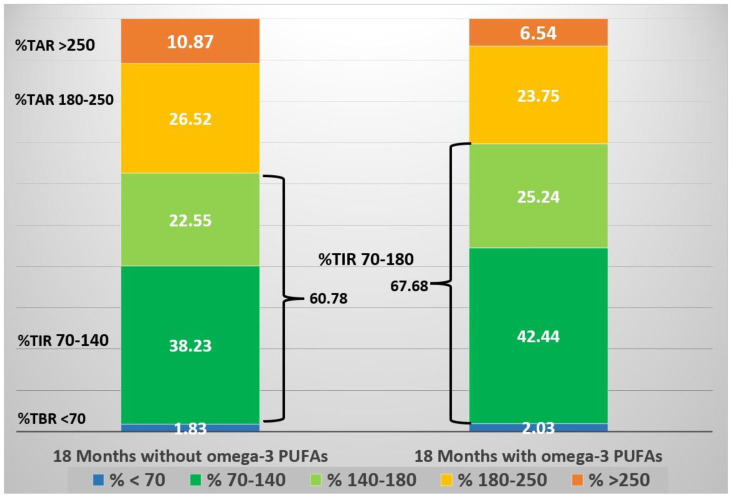
Schematic diagram of CGM metrics before and after the initiation of ω-3 PUFA supplementation. The stacked bars represent the proportion of time (expressed as a percentage) spent within a specific target glucose range [88].

**Figure 4 nutrients-16-03639-f004:**
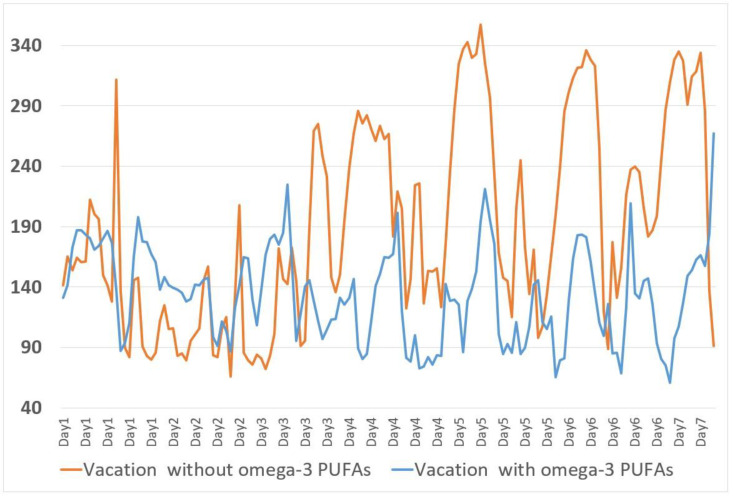
The average subcutaneous glucose levels referring to 2 weeks of mountain vacations with daily comparable physical activity consisting of walking, climbing, and cycling without (orange line) or within (blue line) ω-3 PUFA supplementation periods, indicating a reduction in daily fluctuations in the second [88].

**Figure 5 nutrients-16-03639-f005:**
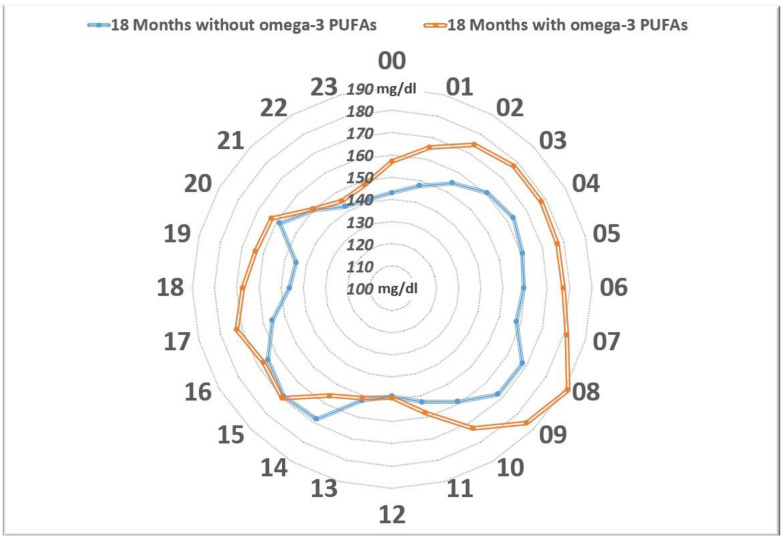
Distribution of the mean subcutaneous glucose levels over 24 h. The orange line represents values during the 18-month-period prior to the initiation of the ω-3 PUFA supplementation, whereas the blue line represents the 18 months following the initiation. The blue line encircles a smaller area than the orange one. From the midnight to the midday constant, lower mean values were observed upon supplementation with ω-3 PUFAs, while the levels appeared to be similar between the two study periods at mealtime (lunch, afternoon snack, and dinner). The rapprochement of the circles in conjunction with meals was interpreted as the effect of boluses before meals. Instead of the fasting periods, in the context of the substantial stable insulin demand of the girl, an increased insulin sensitivity with the ω-3 PUFA supplementation was highlighted [88].

**Table 1 nutrients-16-03639-t001:** Daily nutrient intake from the ISPAD Guidelines 2022 for children and adolescents [41].

Carbohydrates must contribute 40–50% of daily energy intake
Moderate sucrose intake (up to 10% of total energy)
Fat should represent 30–40% of total energy
<10% saturated fat + trans fatty acids
Proteins should represent 15–25% of total energy

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
