# Peer review of "Insights in Nutrition to Optimize Type 1 Diabetes Therapy"

_nutrients, 2024, doi:10.3390/nu16213639_

Round 1

Reviewer 1 Report

Comments and Suggestions for Authors

The article needs a few corrections.

The statement that only breastfeeding during infancy protects against the development of type 1 diabetes is questionable. The author writing this sentence causes only one paper from 2004.  In the following years, there was a large TRIGR study, lasting 12 years, which did not confirm the protective effect of breastfeeding on the development of type 1 diabetes. The author did not refer to this study at all.

CGM is not a control of glucose in the blood, but in the interstitial fluid. 

It is worth referring to the ISPAD recommendations of 2022, not 2018.

Author Response

1° Reviewer

Thanks for the accurate revision.

  1. I have changed in Background (page 1, lines 39-44) the sentence “Only breastfeeding shows evidence on prevention of type 1 diabetes why it should always be recommended” with “Waiting for further studies, at today breastfeeding must be encouraged and early introduction of nutrients must be carefully delayed for the role in determining gut microbiota as possible players in the process trigger of autoimmunity. In Conclusions (page 22, lines 895-897) “Breastfeeding must be recommended, to endorse the superiority of gut microbiome induced by breast milk in the window period of the first years of life, and the early introduction of solid nutrients, gluten and cow milk it must be not recommended [4].

In Reference [4] the article by Sadauskaite-Kuehne, V., Ludvigsson, J., et al. Longer breastfeeding is an independent protective factor against development of type 1 diabetes mellitus in childhood. Diabetes/metabolism research and reviews2004 20(2), 150–157. https://doi.org/10.1002/dmrr.425 was changed with Quinn LM, Wong FS, Narendran P. Environmental Determinants of Type 1 Diabetes: From Association to Proving Causality. Front Immunol. 2021;12:737964. Published 2021 Oct 1. doi:10.3389/fimmu.2021.737964 because if think is more explicative of the whole possible nutritional determinants in type 1 diabetes and of the importance of the gut microbiota, than “Knip M, Akerblom HK, Al Taji E, Becker D, Bruining J, et al. Effect of Hydrolyzed Infant Formula vs Conventional Formula on Risk of Type 1 Diabetes: The TRIGR Randomized Clinical Trial. JAMA (2018) 319(1):38–48. doi: 10.1001/jama.2017.19826” centered on conventional and hydrolyzed infant formula, or “Lampousi AM, Carlsson S, Löfvenborg JE. Dietary factors and risk of islet autoimmunity and type 1 diabetes: a systematic review and meta-analysis. EBioMedicine. 2021;72:103633. doi:10.1016/j.ebiom.2021.103633”.

  1. I have corrected in the whole text CGM as the continuous subcutaneous glucose monitoring.

Reviewer 2 Report

Comments and Suggestions for Authors

Very informative manuscript concerning optimizing nutrition in Type 1 Diabetes according to the state-of-the-art, written by an scientifically very experiened author.

Yet, for me as reader was the paper very hard to read and sometimes hard to understand because of the partly incomprehensible English language which has to be profoundly improved before publication maybe by help of an English native speaker.

Some other points from my side:

1.) Please omit the repetitive word "anyway"

2.) Figures and Tables coud be "optically" improved in design and quality.

3.) Perhpas, parts of the manuscripts could7should be shortened (f.e. the chapter concerning dyslipidaemia where in respecct to cardiovascular-protection only high dose EPA could show vardiovascular benefit in type -2 Diabetes nellitus (REDUCE-IT study in the New Engl J Med and not a mixture of Omega-3 PUFAS), In my knowledge no hard-endpoint data are available for typ 1 diabetes mellitus. Thus lipid-lowering action is often mandatory with pharmacological agennts (i.e. primarily statins) accotding to guidelines (i.e. EAS/ESC lipid guideline 2019 and review "Typ 1 diabetes mellitus" in New Engl J Med 2024).

4.) The "conclusions" at the end of the paper should be put in short-cut in a "message box" at the beginning of the paper to get more attention by the potential readers who want to transfer the presented knowledge into clinical practice.

Comments on the Quality of English Language

Please improve the quality of English language, maybe by helf of an Englsih native speaker or a tranlation bureau. 

Author Response

Thanks for your accurate revision.

  1. I agree. I’ve delated “anyway” in the whole text.
  2. Figures and the table are replaced. Now the figures are reproduced with increased definition and quality.
  3. Two parts of the text are shortened.

The first in chapter “The Low ω-6 to ω-3 polyunsaturated fatty acid ratio diet” (page 9, lines 388-391).

The paragraph "Moreover, ω-3 PUFAs supplementation implies lowering triglycerides, reduction in of cardio-vascular risk, inflammation, decreased intake of ω-6, of relevant interest in NAFLD, obesity, type 2 and 1 diabetes. The optimal AA/EPA range is 2-2.5 [arachidonic acid ω-6 (AA): eicosapentaenoic acid ω-3 (EPA)]." has been removed at all, being not reported in the article cited as the font.

The second as suggested, in chapter “The diet in Type 1 diabetes and Dyslipidemia” (page 11, lines 477-481), the sentence was shortened “Innovative therapies are envisaged, including nutritional ones as purified ω-3 PUFA at pharmacological doses (2-4 g/day) and icosapent ethyl (a highly purified EPA eicosapentaenoic acid ethyl ester) in type 2 diabetes [62,63]. 4g/day) to lower triglyceride levels [REDUCE-IT Clinical Trial, 64] and reduce cardio-vascular associated risk” with “Innovative therapies are envisaged, including nutritional ones as icosapent ethyl (a highly purified EPA) in type 2 diabetes [62,63]. “

  1. A message box was made as graphical abstract.

A revision of the text for English was done.

Thanks very much for corrections and suggestions.

Reviewer 3 Report

Comments and Suggestions for Authors

The review article, "Insights in Nutrition to Optimize Type 1 Diabetes Therapy," provides a comprehensive overview of the role of nutrition in managing Type 1 diabetes across its various stages. While the article thoroughly examines the topic and is well-referenced, it lacks clarity in language, organization, and accessibility, making it difficult for the average reader to understand and interpret. Detailed comments are as follows:

1.        The English used in the article shows significant grammar, syntax, and sentence structure issues. Several sentences are convoluted or contain awkward phrasing, making them difficult to follow. For example, phrases like "interesting was identified a window period" and "So, there is a variable time from the first years of live" are grammatically incorrect and disrupt the reading flow.

2.        The word choice is often incorrect. Terms are sometimes used incorrectly or in a way that can confuse the reader. For instance, phrases like "maximalize outcomes" should be "maximize outcomes," and "hypostasized" instead of "hypothesized" indicate a lack of attention to detail.

3.        The article lacks consistency in terminology and style. There are instances where the same concept is described using different terms, which could confuse the reader. Moreover, the style fluctuates between formal scientific writing and less formal expressions, which could detract from the article’s professionalism

4.        The article is organized into sections that logically follow the progression of Type 1 diabetes from preclinical stages to chronic management. However, the transitions between sections are not always smooth, and the narrative sometimes lacks coherence, making it difficult to focus.

5.        In several instances, the references are inserted into the text in a way that disrupts the natural flow of the narrative. For example, references are sometimes placed in the middle of sentences or clauses, interrupting the reader’s understanding of the main point. This can make the text harder to read and follow. Refer to this line: “The study found that the intake of ω-3 PUFAs can reduce the risk of Type 1 diabetes by modulating the immune response [16], a finding that has significant implications for dietary recommendations."

The article needs editing to improve the language and enhance the logical flow between sections, ensuring that each section transitions smoothly into the next.

Comments on the Quality of English Language

The quality of writing, English language usage ( grammar and spelling), sentence construction, and overall readability need extensive improvement. 

Author Response

Thanks for your revision.

  1. The whole text was revised for English. The sentences (in chapter Preclinical Diabetes, from line 55 to line 57, page, 2,) “interesting was identified a window period” changed “ Environmental events causing type 1 diabetes should be identified within the first two years of age, “ and (in chapter Presentation of Diabetes from line 122 to line 124, page 3) “So there is a variable time from the first years of life” changed “A variable span of time from the first years of live to the clinical onset of the disease peaking at puberty during which the autoimmunity selectively reduces b-cell mass, should be the ideal period to counteract the process.”
  2. Some words have been changed “maximalize” in “maximize” (chapter Clinical Onset…, line 224, page 6) and “hypostatized” in “hypothesized” (chapter The Ketogenic Diet, line 432, page 10).
  3. References are placed now correctly at the end of phrases in the whole text.

Round 2

Reviewer 2 Report

Comments and Suggestions for Authors

Fine revsion

Author Response

Thank you for your review and final judgment. I think it is necessary to send to you the last text of the article after revision according to the indications of the third reviewer, and of the English. I also added an Graphical Abstract as a more attractive alternative to the Message Box, according to your suggestion. Thanks for this too.

Reviewer 3 Report

Comments and Suggestions for Authors

The revised version has made some grammatical improvements, and the text reads slightly more fluidly than the original. However, there are still numerous awkwardly structured sentences and occasional grammatical mistakes

·      The article's organization is generally coherent, with sections that follow the progression of diabetes. However, the transitions between topics remain somewhat abrupt. For instance, the shift from discussing nutritional models to autoimmune triggers in preclinical diabetes feels disjointed, and readers might find it difficult to follow the overall argument.

·      Line 720: Fig.3 “In an adolescent type 1 diabetes girl, along 3 years of disease were recruited 344,020 data of subcutaneous glucose levels equally divided before and after the initiation of ω-3 PUFAs supplement” : Error in sentence construction.

·      Overtly long, complex sentences (624-628) : Bearing in mind that in type 1 diabetes insulin is administered subcutaneously, where it is inadequate to act physiologically (from the endocrine pancreas through the portal vein, to the liver filter, and then through the systemic circulation to the tissues), as basal insulin and mostly in conjunction with meals as boluses, follows that this insurmountable inadequacy must be compensated at least in part, by nutrition limiting postprandial hyperglycemia.

·      Reference placement is incoherent (601-605) : In summary, there is concordance in the literature on some points: (1) nutrition variety promotes the richest diversity of the GM, which reduction accelerates the evolution of type 1 diabetes, (2) SCFAs in the gut show protective effects against early-onset human type 1 diabetes, and (3) higher concentrations of acetate and butyrate SCFAs can be safely produced by administration of amylose-modified corn resistant starch (HAMSAB).

·      Very obvious grammatical mistakes and awkwardly phrased sentences, beginning with “was” or “were” (line 589-593): “ Was administered supplement for 6 weeks with amylose maize-resistant starch modified with acetate and butyrate (HAMSAB) [82]. In a 12 weeks follow-up was assessed changes in the gut microbiota. Were found increased concentrations of SCFA acetate, propionate, and butyrate in stools and plasma, in concert with a shift in the composition of the GM

While I appreciate the author's efforts in revising the manuscript, I regret that several concerns raised in my previous review have not been fully addressed. Although some textual improvements have been made, the article requires significant revision to enhance its clarity. A thorough review of the article for English, grammar, and structure is still necessary to ensure that it meets the standards required for publication.

Comments on the Quality of English Language

The article needs thorough improvement in the quality of English before it can be considered for publication. 

Author Response

Thank for your further precise revision of the manuscript “Insights in Nutrition to optimize Type 1 Diabetes” nutrients-3191950. Your corrections allowed to me to improve the article in the contents before the editing for English by an expert native translator.

1° in Preclinical Diabetes, page 2, lines 72-77, I added a connecting sentence.

There is general accordance that the rising incidence of type 1 diabetes cannot be ascribed to genetics alone, and among causative environmental triggers were considered some related to nutrition or diet. Specifically breastfeeding, cow’s milk, formula milk, gluten, probiotic, vitamin D, nicotinamide, omega-3 were investigated and related to nutrition, birth weight, weight gain, BMI, childhood obesity and gut microbiota [4].

Thanks for the suggestion.

2° in the chapter Diet and gut microbiota in type 1 diabetes, a substitution of phrases was done, page 13, lines 594-606, resulting also a shorter text:

Was administered supplement for 6 weeks with amylose maize-resistant starch modified with acetate and butyrate (HAMSAB) [82]. In a 12 weeks follow-up was assessed changes in the gut microbiota. Were found increased concentrations of SCFA acetate, propionate, and butyrate in stools and plasma, in concert with a shift in the composition of the GM. A diet with supplement for 6 weeks with amylose maize-resistant starch modified with acetate and butyrate (HAMSAB) [82].

A diet with amylose maize-resistant starch modified with acetate and butyrate (HAMSAB) was administered for 6 weeks in 25 patients and a 12-week follow-up was made to evaluate changes in the gut microbiota over time. A shift in the composition of gut microbiota and increased concentrations of SCFA (acetate, propionate, butyrate) in stools and plasma were found.

Thanks for correction.

3° In the same Chapter, page 14, lines 615-624, a change of sentence was done with corrections.

In summary, there is concordance in the literature on some points: (1) nutrition variety promotes the richest diversity of the GM, which reduction accelerates the evolution of type 1 diabetes, (2) SCFAs in the gut show protective effects against early-onset human type 1 diabetes, and (3) higher concentrations of acetate and butyrate SCFAs can be safely produced by administration of amylose-modified corn resistant starch (HAMSAB).

Some deductions seem emerge overall: the gut microbiota play a protective or favoring role in diabetes pathogenesis; the SCFAs in the gut show protective effects against early-onset type 1 diabetes; specific foods can increase the SCFA concentration into the gut [77-82].

Thanks for corrections.

4° In Diet to improve Glycemic Variability in Type 1 Diabetes, page 14, lines 642-652, the sentence was replaced and shortened.

Bearing in mind that in type 1 diabetes insulin is administered subcutaneously, where it is inadequate to act physiologically (from the endocrine pancreas through the portal vein, to the liver filter, and then through the systemic circulation to the tissues), as basal insulin and mostly in conjunction with meals as boluses, follows that this insurmountable inadequacy must be compensated at least in part, by nutrition limiting postprandial hyperglycemia.

Glucose variability at today represents one of most critical point of type 1 diabetes care. Given that subcutaneous insulin administration bypasses the hepatic filter that physiologically halves its concentration, the adequate dosage of boluses at meals often can be addressed with a diet limiting post-prandial hyperglycemia.

Thanks for suggestion.

5° Figure 3, page 17, lines 743-749, the caption was changed and shortened.

In an adolescent type 1 diabetes girl, along 3 years of disease were recruited 344,020 data of subcutaneous glucose levels equally divided before and after the initiation of ω-3 PUFAs supplement (50 mg/Kg/day, EPA 66% DHA 33%). The CGM metrics of 1.5 years before and after supplement were compared. The stacked bars represent the percentage of time spent within specific target of subcutaneous glucose levels.

Schematic diagram of CGM metrics before and after the initiation of ω-3 PUFAs supplementation. The stacked bars represent the proportion of time (expressed as percentage) spent within specific target glucose range.

Thanks for your accurate revision. I hope that this revised manuscript has achieved sufficient clarity for publication.

Round 3

Reviewer 3 Report

Comments and Suggestions for Authors

The authors have considerably improved the manuscript, and most issues have been addressed,  a few problems still remain, and addressing these would make the manuscript suitable for publication: 

1. There are abrupt jumps between topics, especially in sections related to the nutritional impacts of different substances (for example, vitamin D, gluten, omega 3 PUFAs). For example, the shift from discussing preclinical diabetes to glycemic variability feels sudden and authors should add a more straightforward introduction or summary at the end of each section.

2. Some repetitive sections need to be rephrased. Discussions about specific topics, such as the role of omega-3 PUFAs and vitamin D, appear in multiple sections with very similar phrasing. The authors should consolidate the information into fewer focused paragraphs. 

Comments on the Quality of English Language

The quality of English has improved noticeably in this revision; however, the authors are encouraged to proofread thoroughly before submission to ensure clarity and precision. 

Author Response

Thanks again for your corrections, suggestions and support in this work.

I reduced the repetitions between conclusions of the different chapters. However, some repetitions hold again. In the planning of this review, the division into separate chapters was designed to display the topic by a more useful and attractive way for the presumably different readers. In each chapter were distinguished what the current scientific research demonstrates and the possible conclusions, more subjective and stimulating for further research. So, a lower grade of repetitions in conclusions stands again as a connecting item between the different chapters.

Moreover, as suggested by the second Reviewer, a summary was done as Graphical Abstract to resume the main points by an operative way for the improvement of nutrition in type 1 diabetes.

The most of following corrections become from a low-grade re-editing of the text after English corrections, to make its meaning right and easier to be understood. Thank you again for this suggestion.

  • In Background 2, at line 44-45 a correction: … due to their roles in determining gut microbiota, as they are and of possible players in the process of triggering autoimmunity.
  • In Background I’ve changed the sentence in Pag 2, line 48-50, with a more connecting one. Thanks for the suggestion.

Metabolic instability, which represents a daily commitment for dietitians, doctors, and patients, presents a dietary competence that must be addressed in this work.  

Moreover, the control of the glycemic variability, which represents a major commitment for patients, might be reduced through nutrition and diet modifications, so must be addressed in this work.

  • In Clinical onset and presentation, 4, at line 185-186 I’ve delated some words, put brackets and added ‘without’, changing the sense of the sentence:

…. concluded that there were different age-related subtypes (or “endotypes”) of type 1 diabetes with different characteristics, among which the C-peptide levels referred to the age of onset of type 1 diabetes; they were (lower in younger children) and did not have without significant age-related differences in the progressive rate of its decline.

  • In Clinical onset and presentation, 5, line 215-217 the sentence was modified:

In one cohort study, a whole type 1 diabetes series attending a single pediatric service was supplemented with 1000 IU of cholecalciferol plus sea-fish-derived ω-3 PUFAs (1000 IU/day, plus 60 mg/Kg/day) starting within the first trimester of the clinical onset of the disease [37]. A retrospective comparison with a previous series that received only cholecalciferol (1000 IU/day) showed that …

  • In The intake of macronutrients and its effect on post-prandial glycemia, Pag. 8, line 310, a word was changed:

GLP-1 and GIP, but there was an elevation in of both after high-protein and high-fat meals.

  • In Situations and comorbidities 8 line 327 a sentence was added:

The management of diabetic ketoacidosis constitutes an important chapter in the treatment of type 1 diabetes that must be carefully evaluated by clinicians, beyond the scope of this review.

  • In Diabetic ketoacidosis, Pag 8, line 332, a modification was done:

the calibration of insulin delivery and with carbohydrate intake per os or via IV are is crucial.

  • In Mediterranean and low carbohydrate diets, 9, line 376:

… the Mediterranean diet was found to cause gain improvements in HbA1c% …

  • In The diet in Type 1 diabetes and Dyslipidemia, Pag. 12, line 495-497 a reduction of the sentence was done:

and ω-6: ω-3 ratio < 3 improved metabolic control A low AA–EPA (arachidonic acid ω-6 (AA)–eicosapentaenoic acid ω-3 (EPA)) ratio of <3 seems to be a promising indicator of metabolic control

  • In The gluten free diet in type 1 diabetes and Celiac Disease, Pag. 13, line 540, a change was done:

An anecdotal case was reported with of a 6-year-old child

  • In The gluten free diet in type 1 diabetes and Celiac Disease 13, line 561, a change was done:

supported by the a solid scientific literature

  • In The diet and gut microbiota in type 1 diabetes, Pag. 14, line 602:

GM was changed with gut microbiota

  • In The diet and gut microbiota in type 1 diabetes 14, line 627:

and a more circulating B and T cells with the regulatory phenotype.

  • In Diet for improving Glycemic Variability, Pag.15, at line 668, a sentence was delated because a repetition:

Glucose variability today represents one of most critical points of type 1 diabetes care.

  • In Diet for improving Glycemic Variability, 16, line 699, a sentence was changed:

Therefore, every single patient should be seen as a priority, and it is necessary to counteract glycemic instability through diet to reduce GV through dietary components, would be one of diabetologist’s major commitment.
